# Essential roles of plexin-B3+ oligodendrocyte precursor cells in the pathogenesis of Alzheimer's disease

Naomi Nihonmatsu-Kikuchi[1], Xiu-Jun Yu[1,2], Yoshiki Matsuda [1], Nobuyuki Ozawa[1], Taeko Ito[1], Kazuhito Satou[3], Tadashi Kaname [3], Yasushi Iwasaki[4], Akio Akagi[4], Mari Yoshida[4], Shuta Toru[5], Katsuiku Hirokawa[5], Akihiko Takashima[6], Masato Hasegawa[7], Toshiki Uchihara[1,5] & Yoshitaka Tatebayashi [1✉]

The role of oligodendrocyte lineage cells, the largest glial population in the adult central nervous system (CNS), in the pathogenesis of Alzheimer's disease (AD) remains elusive. Here, we developed a culture method for adult oligodendrocyte progenitor cells (aOPCs). Fibroblast growth factor 2 (FGF2) promotes survival and proliferation of NG2+ aOPCs in a serum-free defined medium; a subpopulation (~5%) of plexin-B3+ aOPCs was also found. FGF2 withdrawal decreased NG2+, but increased plexin-B3+ aOPCs and Aβ1-42 secretion. Plexin-B3+ aOPCs were distributed throughout the adult rat brain, although less densely than NG2+ aOPCs. Spreading depolarization induced delayed cortical plexin-B3+ aOPC gliosis in the ipsilateral remote cortex. Furthermore, extracellular Aβ1-42 accumulation was occasionally found around plexin-B3+ aOPCs near the lesions. In AD brains, virtually all cortical SPs were immunostained for plexin-B3, and plexin-B3 levels increased significantly in the Sarkosyl-soluble fractions. These findings suggest that plexin-B3+ aOPCs may play essential roles in AD pathogenesis, as natural Aβ-secreting cells.

[1] Affective Disorders Research Project, Tokyo Metropolitan Institute of Medical Science, Setagaya, Tokyo, Japan. [2] Department of Neurology, Key Laboratory of Neurology of Hebei Province, Second Hospital of Hebei Medical University, Shijiazhuang, China. [3] Department of Genome Medicine, National Center for Child Health and Development, Setagaya, Tokyo, Japan. [4] Institute for Medical Science for Aging, Aichi Medical University, Nagakute, Aichi, Japan. [5] Department of Neurology, Nitobe Memorial Nakano General Hospital, Nakano, Tokyo, Japan. [6] Department of Life Science, Gakushuin University Graduate School of Science, Toshima, Tokyo, Japan. [7] Dementia Research Project, Tokyo Metropolitan Institute of Medical Science, Setagaya, Tokyo, Japan. ✉email: tatebayashi-ys@igakuken.or.jp

The amyloid hypothesis for Alzheimer's disease (AD) posits a neuron-centric, linear cascade initiated by the abnormal production of longer amyloid β peptide (Aβ) forms, especially Aβ1-42, from amyloid precursor protein (APP); this leads progressively to tau pathology, synaptic dysfunction, inflammation, neuronal loss, and ultimately dementia[1]. Activated astrocytes and microglia are commonly found as part of glial nests around senile plaques (SPs) composed of Aβs[2]. Such reactive gliosis occurs as a multicellular neuroinflammatory response to Aβ accumulation and is believed to contribute to the clearance and removal of extracellular Aβs[1] and/or progression of synapse loss by engulfment of synapses[3]. However, while these cells have therefore received much attention[1], the roles played by other major glia, such as oligodendrocyte lineage cells, in the pathogenesis of AD remain largely unknown.

Oligodendrocyte lineage cells constitute ~75% of the neuroglial cells in the neocortex, and are thus the largest group of non-neuronal cells in the adult human brain[4]. While mature oligodendrocytes produce myelin and facilitate neuronal transmission, the roles played by adult oligodendrocyte progenitor cells (aOPCs), which are characterized by the expression of platelet-derived growth factor receptor α subunit (PDGFRα) and nerve/glial antigen-2 (NG2), a proteoglycan core protein[5,6], are still unknown. aOPCs descend from OPCs in the perinatal CNS[7] and are distributed throughout the adult brain, comprising up to 5–10% of adult CNS cells[8]. Although they generally continue to proliferate, some generate myelinating oligodendrocytes in the gray and white matter even during adulthood[8,9]. Furthermore, in the human prefrontal and entorhinal cortex, myelination in the gray matter continues throughout life, peaking around the 5th–6th decades[10,11].

Braak & Braak found a link between neuron vulnerability in AD and cortical (gray matter) myelination, based on the fact that the spreading of neurofibrillary tangles follows the pattern of cortical myelination during adulthood in reverse order[12]. Although this neuropathological finding clearly demonstrates the close relationship between adult gray matter oligodendrocyte lineage cells and AD pathogenesis, it has been underestimated; this is partly due to the lack of appropriate markers and/or in vitro systems to investigate such relationships in humans and rodents. Recently, Mathys et al. investigated transcripts in AD brains at single-cell resolution, and unexpectedly found that early AD stage myelination and axonal integrity and repair may essentially be involved in pathogenesis[13]. Furthermore, Chen et al., using spatial transcriptomics, identified 57 plaque-induced genes in SP microenvironments in AD and found a considerable oligodendrocyte gene response[14]. However, little is known about the changes in oligodendrocyte lineage cells with healthy aging, and their role in the initial development and progression of AD[15].

CNS cell purification and culture from embryonic or postnatal (up to approximately P20) rodent brains are possible using several different protocols; however, those from adult (>2 month old) brains remain more challenging and often result in negligible or low yields, except for a few cell types (i.e., adult microglia and hippocampal neural stem/progenitor cells [NSPCs]) that reside within a highly specific stem cell niche in the dentate gyrus[16–19]. Culturing oligodendrocyte lineage cells from adult brains is also considerably complex. OPCs from postnatal and adult rodent optic nerves can be purified by using a combination of sequential immunopannings[20]; however, advancing age, especially after P50, renders any attempt to culture them in vitro increasingly difficult and in fact nearly impossible[21]. Utilizing surgically resected adult human brains, Antel et al. established a method for purifying and culturing oligodendrocyte lineage cells that can survive for up to 4 weeks in vitro[22,23]. However, the use of serum limits their precise functional analysis and purity.

In the present study, we developed a novel method for purifying aOPCs from the adult rat brain (>4 months old) and culturing them for up to several months in a defined medium. This allowed us to identify plexin-B3 as a novel uncharacterized aOPC marker. Interestingly, functional analyses revealed that these cells secrete massive amounts of Aβs, especially Aβ1-42. We also found that plexin-B3 is a unique delayed marker for cortical gliosis following brain injuries, and under certain pathological conditions, plexin-B3+ aOPCs are associated with extracellular Aβx-42 accumulation in vivo. Furthermore, we showed that almost all the SPs in our AD brain samples were immunoreactive for plexin-B3. These data suggest that plexin-B3+ aOPCs play some roles in the pathogenesis of AD, most likely as natural Aβ-secreting cells.

## Results

### Purification and culture of aOPCs from the adult rat brain.
Density gradients have often been used to purify microglia and hippocampal NSPCs from adult rodent brains (gradients spanning 1.030–1.065 g/ml in the case of microglia[16–18] and 1.065–1.074 g/ml in the case of NSPCs)[17–19], as they allow less buoyant cells to be separated)[19]. In these purification procedures, the higher buoyancy (<1.04 g/ml) cell fractions are discarded, because they are regarded as consisting merely of accumulated debris, mainly myelin[16,17,19].

However, our previous studies[24] indicated that many unidentified cell types were present in these more buoyant fractions (<1.04 g/ml), especially those in the density gradient spanning 1.029–1.041 (Fig. 1a).

Preliminary immunolabeling studies revealed that most cells in this fraction were positive for olig2, a transcriptional factor and specific marker for oligodendrocyte lineage cells, while a smaller population was positive for Iba1, a microglial marker. To further purify the olig2+ cells, we overlaid these fractions on poly-D-lysine-coated dishes for 30–60 min and then gently washed out the suspension. As olig2+ cells adhered to the dishes more quickly and tightly, most likely due to their membrane charge, most microglia and debris were removed in the wash suspensions (SW in Fig. 1a).

Some of the attached cells began to proliferate in Neurobasal A/B27 supplemented with FGF2 (Fig. 1b). After 5 days in vitro (DIV), more than 90% of the cells were positively co-immunolabeled with antibodies against NG2, PDGFRα, and olig2 (Fig. 1c).

Only in the primary cultures was a mitogenic effect also found for PDGFaa, although not for epidermal growth factor, nerve growth factor, neurotrophin-3, or ciliary neurotrophic factor. The effects of PDGFaa on cell morphology and migration were different from those of FGF2 (Supplementary Fig. 1). However, after the first passage, aOPC numbers continuously increased only in the medium with FGF2 (Fig. 1d). Within a few passages in FGF2, the cultures became more homogeneous; in fact, nearly 100% of the cells were olig2+ and more than 95% of those were NG2+.

Western blot analysis revealed strong expression of OPC markers, including NG2, PDGFRα, olig2, and Sox10 (Fig. 1e). Markers for mature oligodendrocytes, astrocytes, or neurons remained undetected or detectable only at negligible levels (Fig. 1e). However, in older cultures, such as passage 11 (P11 in Fig. 1e), the levels of some protein markers increased (Sox10, Tuj1) or decreased (NG2) indicating changes in culture properties, most like due to increased heterogeneity. We therefore only employed cultures up to P7 in the subsequent in vitro experiments.

Microarray (Fig. 1f) and RNA sequencing (RNA-seq) (Supplementary Fig. 4A) analyses further confirmed that the cultured

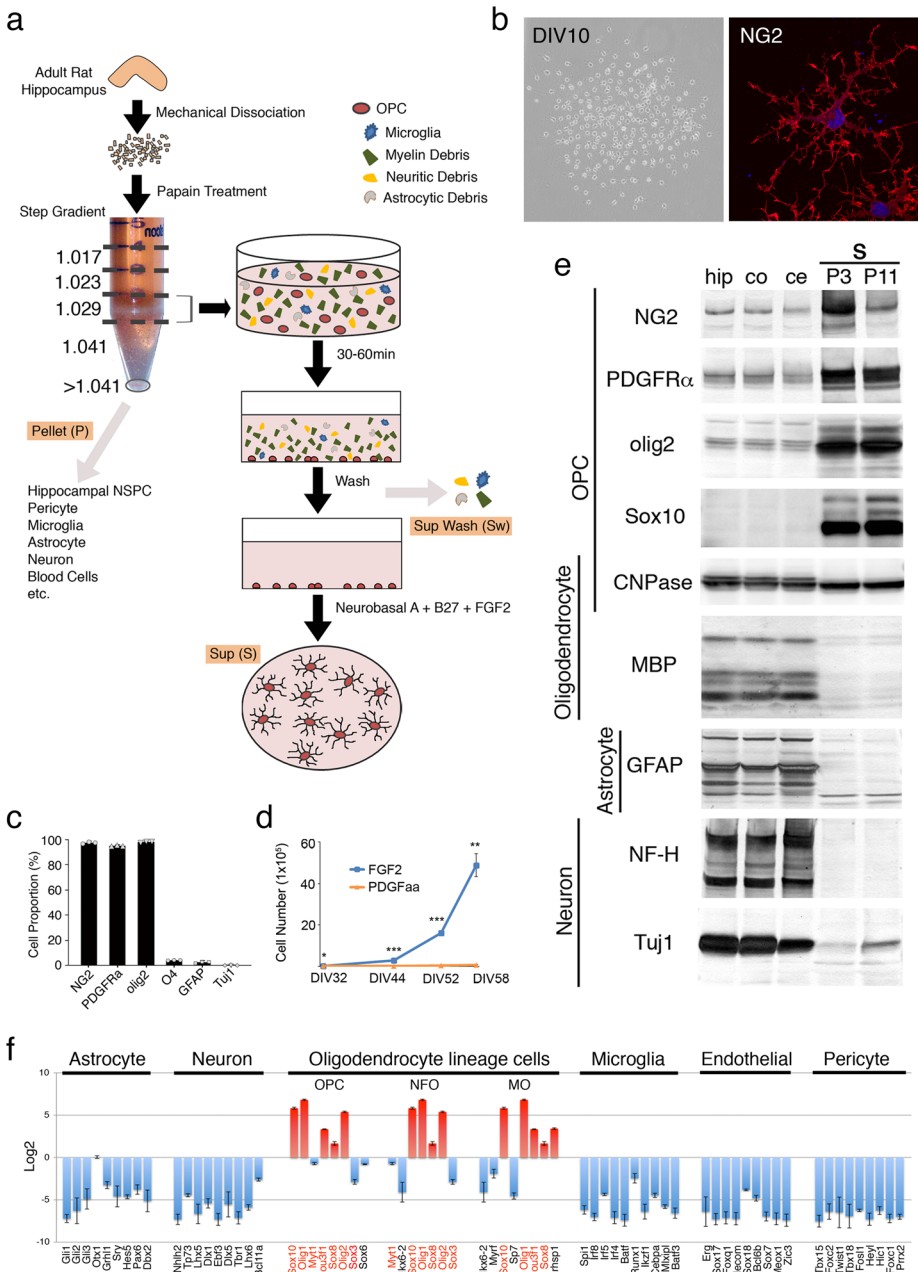

**Fig. 1 Purification and culture of aOPCs. a** Culture strategy. NSPCs: neural stem or precursor cells. **b** Left: Phase contrast images of primary cultures at 10 days in vitro (DIV10). Right: Morphology of an NG2+ cell at DIV10. **c** Proportions (%) of marker-positive cells at DIV 5 (>1800 cells were counted for each marker, $n = 3$). **d** Effects of FGF2 and PDGFaa on cell growth after passage 1 ($n = 3$; ***$P < 0.0001$, **$P < 0.001$, *$P < 0.05$; unpaired t-test. Two-way repeated measure ANOVA also showed a significant difference between the groups [$F_{1, 16} = 4.49$, $P = 0.1 \times 10^{-12}$]). **e** Cell-type specific protein expression profiles of cultured aOPCs and adult rat hippocampus (hip), cortex (co), and cerebellum (ce) cells (20 µg/lane). P3, P11 aOPCs at passage 3 and passage 11, respectively, CNPase 2′,3′-cyclic-nucleotide 3′-phosphodiesterase, PLP proteolipid protein, MBP myelin basic protein, MAG myelin-associated glycoprotein, GFAP glial fibrillary acidic protein, NF-H neurofilament H, Tuj1 neuron-specific class III β-tubulin. **f** Microarray profiles of transcription factor genes in cultured aOPCs. The CNS cell-specific transcription factor genes were those reported by Zhang et al.[25]. Red bars: highly expressed genes; blue bars: downregulated genes; red font genes: overlapped genes between OPC, newly formed oligodendrocyte (NFO), and myelinating oligodendrocyte (MO). See also Supplementary Data 1.

cells expressed oligodendrocyte lineage (especially OPC)-specific transcription factor genes such as *Sox10*, *Olig1*, *Pou3f1*, *Sox8*, and *Olig2*[25]. Comparison of the microarray (Supplementary Fig. 2) and RNA-seq (Supplementary Fig. 3) profiles of cultured aOPCs with two RNA-seq based single-cell transcriptome databases of adult mouse CNS cells (top 40; Zhang et al., 2014[25] and top 50; Wu et al., 2017[26]) further confirmed the relatively specific expression of OPC genes in the cultured cells. Taken together,

these data not only suggest that oligodendrocyte lineage cells, especially aOPCs, can be successfully isolated and cultured from the adult rat hippocampus, but also that FGF2 can effectively maintain aOPC properties for at least several months in vitro.

**Plexin-B3+ aOPCs in vitro**. In the course of our microarray and RNA-seq analyses, we found that the transmembrane protein plexin-B3[27] was highly expressed in cultured aOPCs, and this was

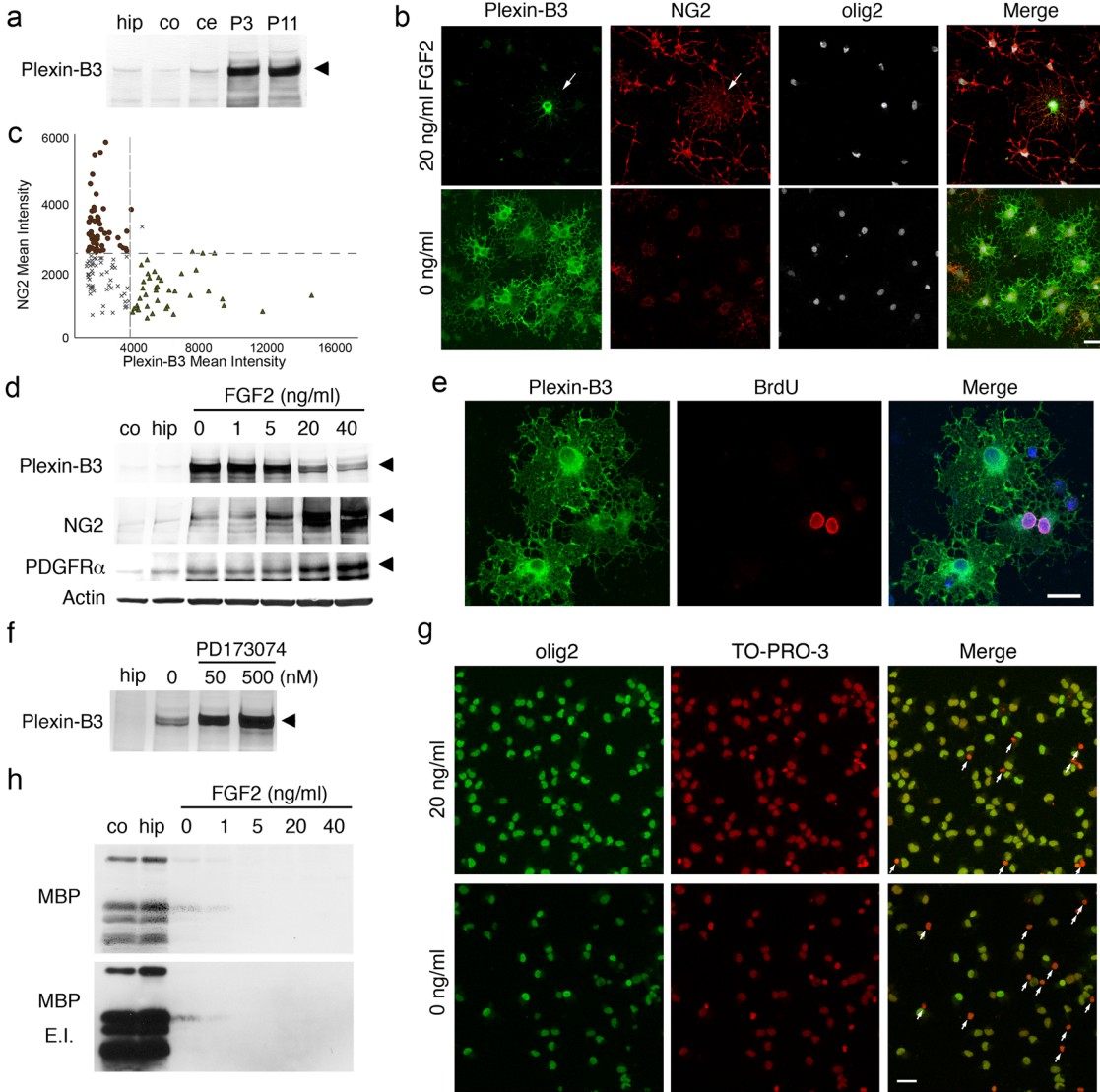

**Fig. 2 Characterization of cultured plexin-B3 + aOPCs. a** Plexin-B3 levels in cultured aOPCs (20 μg/lane). The abbreviations are the same as used in Fig. 1e). **b** Immunocytochemical analysis of aOPCs cultured in medium containing 0 or 20 ng/ml FGF2 for 5 days. Arrows indicate weakly NG2+ plexin-B3+ aOPCs. Scale bar: 20 μm. **c** Quantitative measurements of the intensities of plexin-B3 and NG2 immunoreactive areas of aOPCs cultured in medium containing 0 ng/ml FGF2 for 5 days. An example of a region of interest for a single cell is shown in the left panels of Fig. 3f. Note that plexin-B3+ aOPCs were generally NG2-negative and vice versa. Mean intensity: the integrated total value of signal intensities adjusted by single cell area. n = 172. **d** Effects of FGF2 on the levels of plexin-B3, NG2, PDGFRα and actin (20 μg/lane). For quantification, see Supplementary Fig. 4A. **e** BrdU incorporation in plexin-B3+ aOPCs. Scale bar: 20 μm. **f** Effects of treatment with the FGF inhibitor PD173074 (0, 50, or 500 nM in the presence of 20 ng/ml FGF2 for 24 h; Merck, Darmstadt, Germany), on the levels of plexin-B3. For quantification, see Supplementary Fig. 4B. **g** Effect of FGF2 withdrawal on the proportions of olig2+ cells. White arrows indicate dead cells that were strongly TO-PRO-3 positive. More than 99% of living cells were olig2+ under both conditions (see also Supplementary Table 2). **h** Effect of FGF2 withdrawal on MBP levels. FGF2 withdrawal for 5 days was not enough to increase MBP protein expression, although mRNA levels increased approximately 10-fold (Supplementary Fig. 4D & Data 5). E.I. enhanced image.

further confirmed by western blot analysis (Fig. 2a). Although plexin-family members are generally expressed in neuronal cells, a transcriptome database study by Zhang et al. reported that the plexin-B3 gene was enriched in oligodendrocyte lineage cells, especially in newly formed oligodendrocytes (NFOs)[25]. In our study, immunocytochemical analysis revealed that plexin-B3+ cells comprised less than 5% of total cells but all intensely olig2+ cells (Fig. 2b). Plexin-B3+/olig2+ cells were generally NG2− or weakly NG2+ (Fig. 2b, arrows in the upper panels, Fig. 2c). Moreover, BrdU incorporation studies also revealed that they proliferated in culture (Fig. 2e), suggesting that plexin-B3 is most likely a novel aOPC marker.

Since mitogen withdrawal induces oligodendrocyte differentia-tion of cultured perinatal OPCs[28], the effect of FGF2 withdrawal on the expression of the aOPC marker was studied. Subjecting cultured aOPCs to FGF2 withdrawal for 5 days dramatically increased the proportion of plexin-B3+ aOPCs (29.5 ± 6.9%) and decreased the proportion of NG2+ aOPCs (27.9 ± 3.4%) without changing the proportions of olig2+ cells (>99%) (Fig. 2b, c, g, and Supplementary Table 2). Moreover, plexin-B3+/olig2+ cells remained generally NG2− or very weakly NG2+ (for quantifica-tion, see Fig. 2c). FGF2 withdrawal also induced uniquely ramified and occasionally cored SP-like morphological changes (Fig. 2b and 3e). Western blot analysis further confirmed that

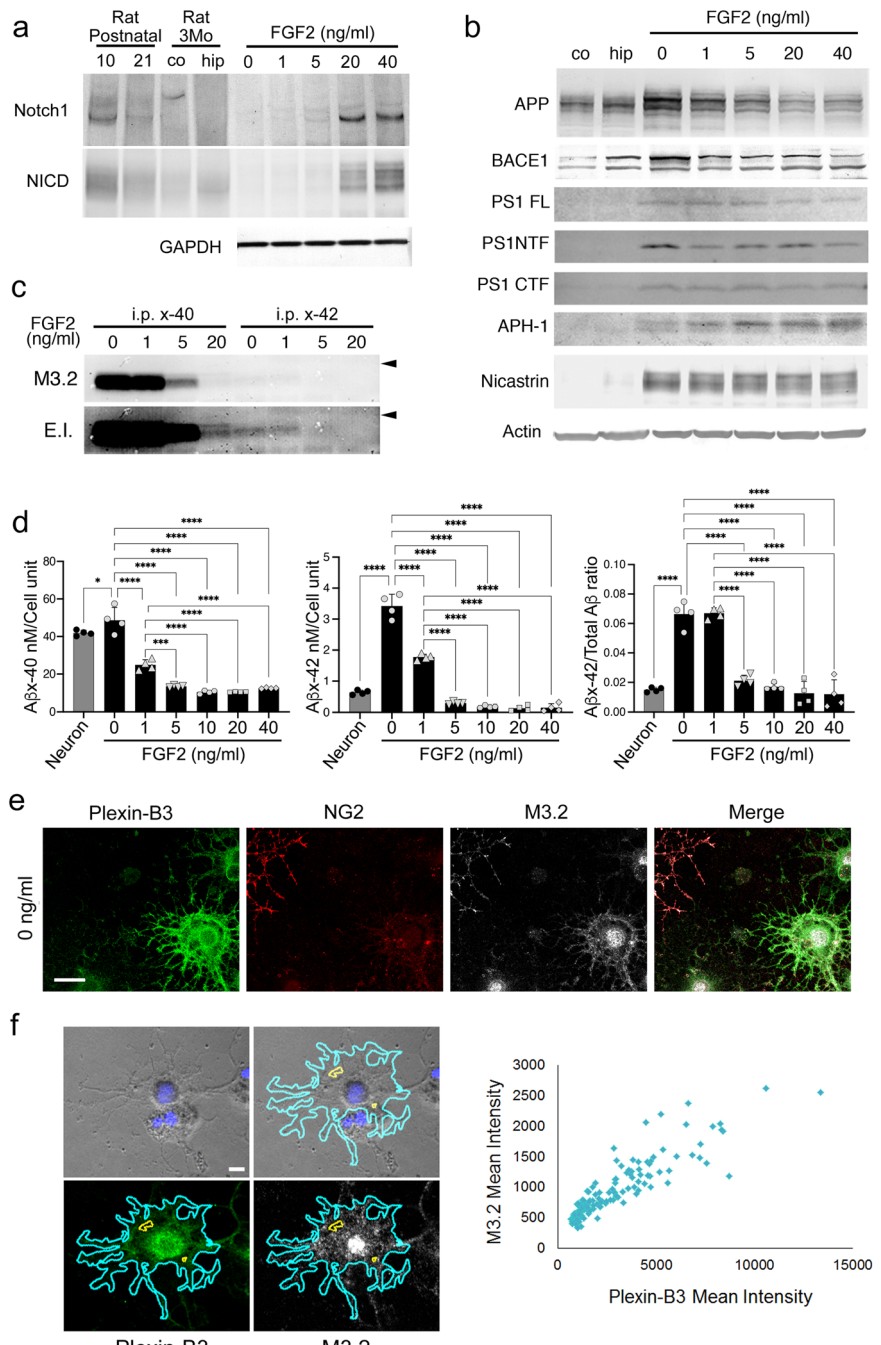

**Fig. 3 Notch and APP processing in cultured aOPCs. a** Effects of FGF2 on the levels of Notch1 and NICDs in cultured aOPCs (15 µg/lane). The expression levels were also compared with those in postnatal rat brains (100 µg/lane) at day 10 and 21 and the adult rat cortex (co) and hippocampus (hip) (3-month old, 100 µg/lane). GAPDH: glyceraldehyde-3- phosphate dehydrogenase. For quantification, see Supplementary Fig. 5A, B. **b** Effects of FGF2 on the levels of APP and β- and γ-secretases (20 µg/lane). BACE1 β-site amyloid precursor cleaving enzyme 1 (β-secretase), PS1 presenilin 1, PS1 FL PS1 full length, PS1 NTF PS1 N-terminal fragment, PS1 CTF PS1 C-terminal fragment, APH-1 anterior pharynx-defective 1. For quantification, see Supplementary Fig. 5C–I. **c** Effects of FGF2 on the levels of secreted Aβs. Western blot analysis of immunoprecipitated (i.p.) Aβs (x-40, x-42). M3.2 monoclonal antibody against rat Aβs (Aβ10-15), E.I. enhanced image, Arrowheads indicate the position of molecular weight marker, 10 kDa. **d** ELISA for rodent Aβx-40 and Aβx-42, and the Aβ42/total Aβ ratio. Secreted Aβx-40 and x-42 levels from cultured fetal rat hippocampal neurons (3 weeks in vitro) are also analyzed ($n = 4$/group; ****$P < 0.0001$, ***$P < 0.001$, **$P < 0.01$, *$P < 0.05$; one-way ANOVA, Tukey's multiple comparisons test [Aβx-40: $F_{6, 21} = 123.2$, $P < 0.0001$; Aβx-42: $F_{6, 21} = 233.9$, $P < 0.0001$; Aβx-42/ Total Aβ ratio: $F_{6, 21} = 65.3$, $P < 0.0001$]). **e** Immunocytochemical analysis of a typical cored-SP-like plexin-b3+ aOPC cultured in 0 ng/ml FGF2 for 5 days. Note that plexin-B3+ aOPC is also M3.2 positive. Scale bar: 15 µm. **f** Quantitative analysis of plexin-B3 and M3.2 immunoreactive areas. An example of region of interest for a single cell is shown in the left panels. Note that as plexin-B3 immunoreactivities increased, M3.2 immunoreactivities increased (right graph). Mean intensity: the integrated total value of signal intensities adjusted by single cell area. Pearson's correlation coefficient: $r = 0.908$, $P < 0.0001$, $n = 148$.

FGF2 withdrawal or administration of the FGF2 inhibitor PD173074 dose-dependently increased plexin-B3 levels (Fig. 2d, f and Supplementary Fig. 4A, B), whereas FGF2 withdrawal decreased NG2 and PDGFRα levels (Fig. 2d and Supplementary Fig. 4A).

We found that the 5-day FGF2 withdrawal did not significantly change the transcription factor gene expression profiles of cultured aOPCs (Supplementary Fig. 4C). In more detailed analyses using the previously described oligodendrocyte lineage-specific gene lists[25,26], we found that 5-day FGF2 withdrawal increased the expression levels of several NFO-, myelinating oligodendrocyte-, and oligodendrocyte-specific genes (Supplementary Fig. 4D, E); however, many OPC genes (such as Gpr17) were still highly expressed in cultured aOPCs.

Interestingly, the levels of myelin basic protein (MBP), an essential marker for mature oligodendrocytes and myelin, increased only faintly (Fig. 2h); however, its mRNA levels increased to extremely high levels after 5-day FGF2 withdrawal (Supplementary Fig. 4D). This phenomenon may represent a type of post-translational repression of MBP, since its protein synthesis is strictly regulated by the temporal and spatial transport of its mRNA and the localized translation of the mRNA into the protein[29,30].

In conjunction, these results indicate that FGF2 withdrawal does initiate oligodendrocyte differentiation of cultured aOPCs, albeit incompletely in vitro; they also suggest that plexin-B3 is most likely an uncharacterized late OPC marker.

**Notch and APP processing in plexin-B3+ aOPCs in vitro.** Since Notch signaling is known to inhibit oligodendrocyte differentiation and myelination[31] of cultured perinatal OPCs, we then studied the effects of FGF2 on Notch signaling in cultured aOPCs. Western blot analysis revealed that FGF2 dose-dependently increased the levels of Notch1, as well as its intracellular signal mediators Notch1 intracellular domains (NICDs), generated by γ-secretase (Fig. 3a, Supplementary Fig. 5A, B)[31]; this suggests that cultured aOPCs do possess regulatory systems for Notch and γ-secretase. FGF2 also increased Notch signaling in a dose-dependent manner, supporting the notion that Notch signaling might play an inhibitory role in aOPC differentiation.

We then investigated the in vitro processing of APP, another major substrate for γ-secretase. APP was also expressed in cultured aOPCs, but in striking contrast to Notch1, FGF2 withdrawal increased its expression (Fig. 3b and Supplementary Fig. 5C). The levels of APP protein in aOPCs cultured in 0 ng/ml FGF2 were higher than those in adult rat hippocampus and cortex tissues (Fig. 3b). The β-secretase BACE1[32] and components of γ-secretase, including presenilin-1[33,34], nicastrin[35], and APH-1[36,37], were also highly expressed in aOPCs cultured at various FGF2 concentrations (Fig. 3b); some of their expression levels were also affected by FGF2 (Fig. 3b and Supplementary Fig. 5D–I). Interestingly, the western blot band patterns for APH-1 and nicastrin changed depending on the concentration of FGF2 (Fig. 3b).

We then studied the secretion of Aβs. Aβ1-40 or Aβ1-42 peptides were immunoprecipitated from the conditioned medium using specific antibodies against the carboxyl (C-) terminal of Aβ1-40 or Aβ1-42, respectively, and analyzed by western blot using the M3.2 antibody, which recognizes rodent Aβ10-15; as the control, secreted APPs were immunoprecipitated using the monoclonal antibody 22c11 (Supplementary Fig. 6A, B). In contrast to those of NICDs, the levels of secreted Aβ1-40 and -42 were most abundant in the medium with 0 ng/ml FGF2 (Fig. 3c), even after adjusting for the secreted APP levels; this suggested that γ-secretase also cleaved APP in

cultured aOPCs, albeit in a manner completely opposite to that of Notch1 processing.

To quantitatively measure the levels of Aβ in the medium, we performed enzyme-linked immunosorbent assays (ELISAs) for Aβx-40 or Aβx-42; a lactate dehydrogenase assay was performed in parallel to adjust for cell number. To compare Aβ levels between cultured aOPCs and neurons, primary fetal rat hippocampal neurons were also cultured (for 3 weeks in vitro). This analysis also confirmed that the levels of both, Aβx-40 and -42 increased as FGF2 concentrations decreased (Fig. 3d). Notably, aOPCs cultured in medium with 0 or 1 ng/ml FGF2 secreted more Aβx-42 than cultured fetal rat neurons, resulting in ~4-fold higher ratios of Aβx-42 to total Aβ in aOPC cultures than in fetal rat neuron cultures (4.39 and 4.45-fold increases, respectively) (Fig. 3d).

To investigate the underlying mechanism of the observed APP processing, we studied the effects of FGF2 on the levels of C-terminal fragments of APP (APP-CTFs) using western blot. APP-C83 and APP-C89 (or phosphorylated APP-C83), two major non-amyloidogenic substrates of γ-secretase, accumulated in aOPCs cultured in 0 ng/ml FGF2; these accumulations gradually diminished when FGF2 levels increased (Supplementary Fig. 6C, D). In striking contrast, APP-C99, the amyloidogenic substrate of γ-secretase that yields Aβs, was barely detectable at lower concentrations of FGF2, but became more detectable as FGF2 levels increased (Supplementary Fig. 6C, D). One of the interpretations of these findings may be the substrate preference of γ-secretase, as indicated by the bands shifts of APH-1 and nicastrin depending on the concentrations of FGF2 (Fig. 3b); γ-secretase may prefer APP-C83/APP-C89 at higher concentrations of FGF2, while preferring APP-C99 (or rejecting APP-C83/APP-C89) at lower FGF2 concentrations, especially 0 ng/ml FGF2 (Supplementary Fig. 6F). This distinctive processing of APP-CTFs, with one dependent on the concentration of FGF2, was not observed in fetal rat neurons (Supplementary Fig. 6C).

We thus examined the levels of secreted p3 (Aβ17-40), a non-amyloidogenic γ-secretase product of APP-C83[38]. Immunoprecipitated Aβx-40 was carefully analyzed by western blot using the antibody against C-terminal Aβx-40. p3-like bands were found immediately beneath those of Aβ1-40, especially under prolonged exposure (Supplementary Fig. 6E; enhanced image, arrowhead). These bands were not detected using the M3.2 antibody that recognizes Aβ10-15 (Fig. 3c, enhanced image), indicating that the detected protein was most likely p3 (Aβ17-40). Interestingly, unlike Aβ1-40 levels, p3-like peptide levels increased when FGF2 increased, supporting the idea that γ-secretase prefers APP-C83 at higher concentrations of FGF2 (Supplementary Fig. 6F).

Immunocytochemical analysis confirmed that cored SP-like plexin-B3+ aOPCs in 0 ng/ml FGF2 expressed APP (Fig. 3e). Quantitative measurements of plexin-B3 and rat APP (using the M3.2 antibody) levels per single cell further indicated that as plexin-B3 levels increased, APP levels increased (Fig. 3f). Since almost all of the plexin-B3+ cells were olig2+ (Fig. 2c, g), we conclude that cultured plexin-B3+ aOPCs possess the ability to secrete considerable amounts of Aβs naturally (without genetic engineering, i.e., the overexpression of substrates and/or secretases with/without point mutations) via totally different processing mechanisms from those reported previously.

**Plexin-B3+ aOPCs in rat brains.** In vivo, plexin-B3+ aOPCs were found to be distributed throughout the adult rat brain (Fig. 4a, b and Supplementary Figs. 7 and 8). In the cortex and hippocampus, anti-plexin-B3 antibodies mostly stained cells with aOPC-like morphologies (Fig. 4a and Supplementary Fig. 8). In the corpus callosum

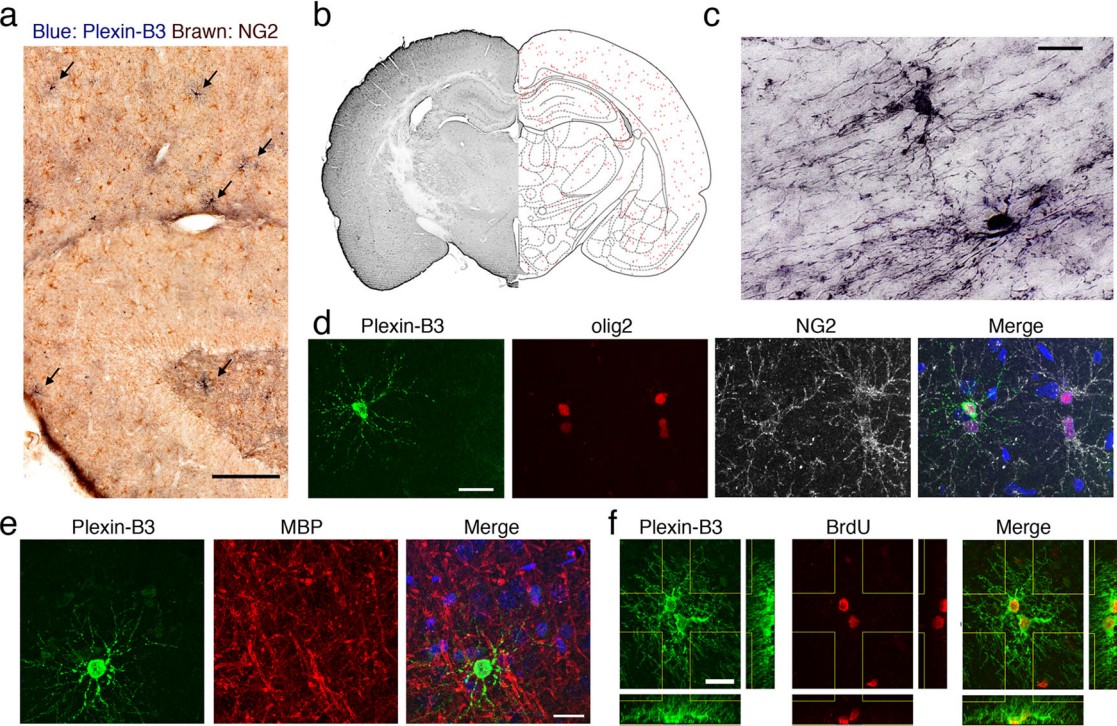

**Fig. 4 Characterization of plexin-B3 + aOPCs in vivo. a** Immunohistochemical analysis of plexin-B3 (blue) and NG2 (brown) in the adult rat hippocampus. Arrows: plexin-B3$^+$ aOPCs. Scale bar: 200 µm. **b** Distribution of plexin-B3$^+$ aOPCs in an adult rat brain section (red dots). **c** Morphologies of plexin-B3$^+$ aOPCs in the corpus callosum. Scale bar: 20 µm. **d** Fluorescent immunostaining of plexin-B3, olig2, and NG2 in the cortex. Scale bar: 20 µm. **e** Fluorescent immunostaining of plexin-B3 and MBP in the cortex. In **d** and **e**, nuclear staining (blue) is shown in the merged panels. Scale bar: 20 µm. **f** Fluorescent immunostaining of plexin-B3 and BrdU in the cortex. Scale bar: 20 µm.

(CC), however, they sometimes stained pre-oligodendrocyte-like cells (Fig. 4c and Supplementary Fig. 7), suggesting the possibility that plexin-B3 may continue to be expressed in the early phase of oligodendrocyte differentiation in vivo.

The plexin-B3$^+$ aOPCs were generally NG2$^-$ (Fig. 4d and Supplementary Fig. 7), glial fibrillary acidic protein (GFAP)$^-$ (Supplementary Fig. 8A), Iba1$^-$ (Supplementary Fig. 8A), MBP$^-$ (Fig. 4e and Supplementary Fig. 7), but they were all olig2$^+$ (Fig. 4d and Supplementary Fig. 7); this further supports the contention that they mostly constitute aOPCs (or early phase of pre-oligodendrocytes in the CC). The density of plexin-B3$^+$ aOPCs in the normal adult rat cortex (~100/mm2) was much lower than those of olig2$^+$ (~1200/mm2) or NG2$^+$ (~400/mm2) aOPCs. Plexin-B3$^+$ aOPCs were still proliferative in vivo, and pairs of plexin-B3$^+$/BrdU$^+$ aOPCs, as well as occasionally those of plexin-B3$^+$/BrdU$^+$ and NG2$^+$/BrdU$^+$ aOPCs, were observed (Fig. 4f and Supplementary Fig. 8B).

**Brain injuries and plexin-B3$^+$ aOPCs.** It is well known that NG2$^+$ aOPCs respond very quickly to brain injuries[39–42], a major risk factor for AD[43]. To investigate the effects of brain injury on plexin-B3$^+$ aOPCs, we first employed the stab wound model. At 2–3 days post stab wound, a dramatic response in NG2$^+$ aOPCs was observed in and around the stab lesions in terms of increased NG2 immunoreactivity as well as hypertrophic morphological changes; interestingly, no such clear plexin-B3$^+$ aOPC response was noted in the same lesions (Fig. 5a, 2Days).

At 6 to 7 days post the stab wound, however, a dramatic increase was observed in the numbers of plexin-B3$^+$ aOPCs. Plexin-B3 was intensely expressed in olig2$^+$ aOPCs in the stab wound wall, which became hypertrophic and formed glial scars (Fig. 5a). In the gray matter near these glial scars, the densities of

plexin-B3$^+$ aOPCs had increased significantly, both laterally and medially (Fig. 5b and Supplementary Fig. 9A).

We then employed the KCl injury model. Topical application of 3 M KCl for 10 min induced cortical spreading depression (CSD) within the ipsilateral but not the contralateral cortex (Fig. 5c, d). CSD is a self-propagating wave of cellular depolarization, that has been implicated as a fundamental mechanism of progressive cortical injury observed in stroke and head trauma[44–46]. At 2–3 days post-KCl application, increased plexin-B3 immunoreactivity was found only within the CC, immediately beneath the necrotic KCl lesions (Fig. 5c, pink-colored area in 2D); however, these white matter reactions decreased to normal levels within 6–7 days.

However, at 6–7 days post-KCl application, plexin-B3$^+$ aOPCs increased in both, around necrotic cortical lesions and the remote ipsilateral cortex (Fig. 5c, e). This delayed cortical gliosis was never induced in the contralateral cortex (Fig. 5c, e). NaCl application did not induce such gliosis either (Fig. 5e); however, slight microgliosis, but not NG2$^+$ aOPC gliosis or astrocytosis, was observed in the same remote ipsilateral cortical areas in this mild CSD model (Fig. 5f and Supplementary Fig. 9B).

One of the unanswered questions is whether plexin-B3$^+$ aOPCs secrete Aβs in vivo. Double immunostaining with the antibodies specific for plexin-B3 and the C-terminal of Aβ1-42 revealed that Aβ1-42-positive dot-like structures were occasionally found in association with plexin-B3$^+$ aOPCs located near the lesions only in the KCl injured brains at 7 days post-injury (Fig. 5g, h). No such structure was found in the rest of the plexin-B3$^+$ aOPCs or in plexin-B3$^+$ aOPCs in the other injury conditions. This finding suggests the possibility that plexin-B3$^+$ aOPCs may secrete Aβs in vivo, but that abnormal Aβ accumulation requires additional mechanisms such as abnormal Aβ clearance[1].

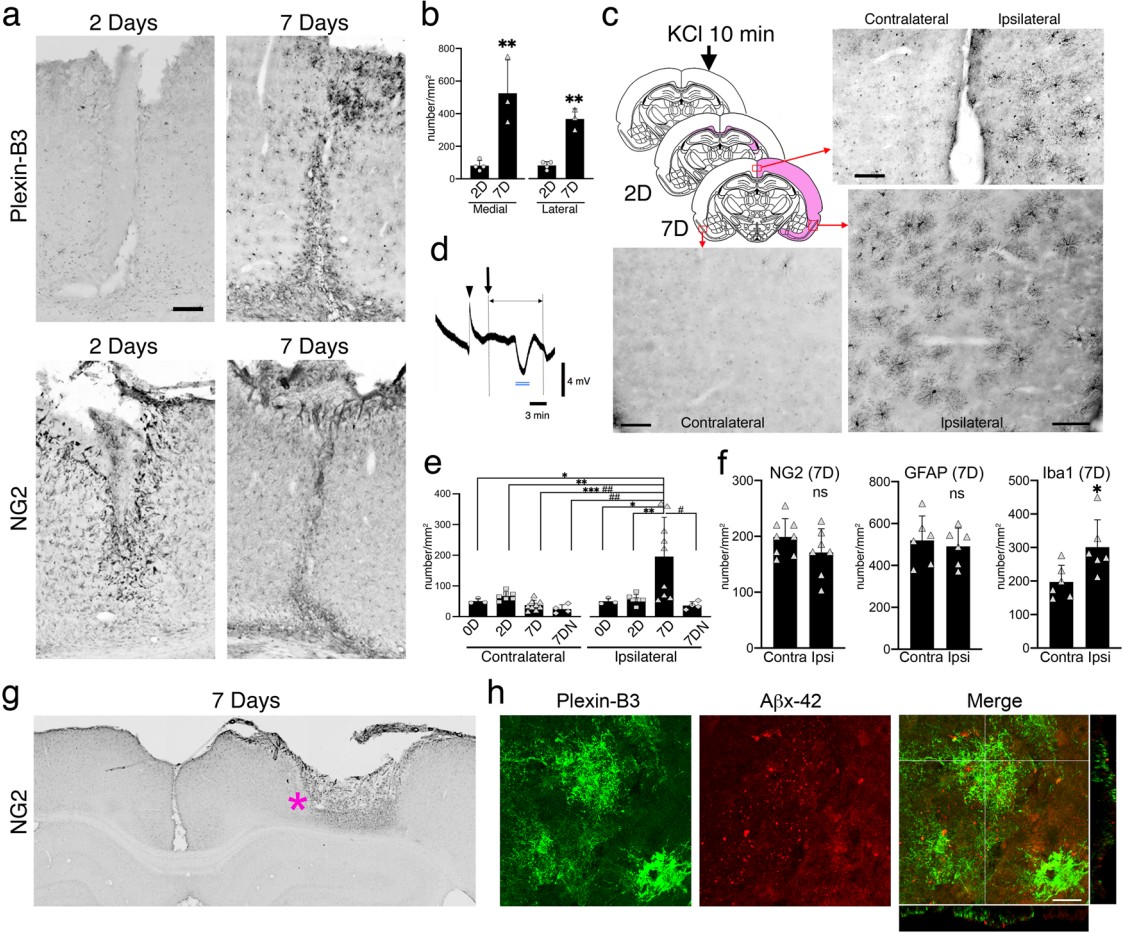

**Fig. 5 Brain injuries and plexin-B3 + aOPCs. a** Stab wound analysis. Plexin-B3 and NG2 immunohistochemical analysis at 2 (2D) and 7 (7D) days after the stab wound. Scale bar: 200 μm. **b** Quantitative cell counts in the defined areas (see Supplementary Fig. 9A) after the injury ($n = 3$, **$P < 0.01$, unpaired $t$ test). **c** KCl injury (arrow), increased plexin-B3 immunoreactive areas (pink coloring) at 2D and 7D, and typical plexin-B3 immunostaining at 7D (from 3 brain areas marked by red rectangles). Scale bar: 100 μm. **d** Representative tracings of direct current potential (millivolts) recorded simultaneously during the 20 min before and after KCl application (arrow). Arrowhead: microelectrode insertion site; blue bars: CSD. **e** Quantification of plexin-B3+ aOPC numbers in the defined areas (see Supplementary Fig. 9B) at 0D, 2D, and 7D after KCl application ($n = 3$–9/group; ***$P < 0.001$, **$P < 0.01$, *$P < 0.05$; two-way (Side x Day) ANOVA, Tukey's multiple comparisons test [$F (2, 32) = 7.701$, $P = 0.0019$]). 7DN: 7D after NaCl application ($n = 4$–9/group; ##$P < 0.01$, #$P < 0.05$; two-way (Side x Treatment) ANOVA, Tukey's multiple comparisons test [$F (1, 22) = 4.913$, $P = 0.0373$]). **f** Numbers of NG2+ aOPCs, GFAP+ astrocytes, and Iba1+ microglia at 7D ($n = 6$–8/group; for Iba1, *$P = 0.024$; unpaired $t$-test; ns not significant). **g**, **h** Extracellular detection of dot-like Aβx-42+ structures outside the few plexin-B3 + aOPCs in the KCl injury model at 7D. **g** Position of the plexin-B3+ aOPCs (pink asterisk). **h** Fluorescent immunostaining using antibodies against plexin-B3 and Aβx-42. Scale bar: 20 μm. Note that all the dot-like Aβx-42+ structures were located extracellularly.

**Plexin-B3 expression in AD brains.** Our in vitro and in vivo findings led us to consider a new idea, that plexin-B3+ aOPCs constitute one of the natural Aβ-secreting cells in AD. To test this idea more directly, we stained paraffin-embedded human brain sections from patients with AD and normal controls (Supplementary Table 3A). In normal control paraffin sections, we could not observe any clear plexin-B3 immunoreactive structures. In AD brain sections, however, anti-plexin-B3 antibodies stained almost all SPs, including the cored ones (Fig. 6a, b and Supplementary Figs. 10, 11 and 12). These plexin-B3+ structures were specific (Fig. 6c, d and Supplementary Fig. 11A), and were co-immunolabeled with antibodies against total Aβ (4G8) as well as Aβ1-42 (Fig. 6e–h and Supplementary Fig. 12). Interestingly, Aβ+ areas of cored SPs were always slightly larger than the corresponding plexin-B3+ cell body areas (Fig. 6e–h, merge, see also Supplementary Fig. 12), indicating extracellular precipitation of Aβs. We also confirmed that these plexin-B3+ SPs were closely associated with, but clearly distinct from, microglia and astrocytes (Fig. 6i–l). The cores of some plexin-B3+ SPs were found to be

immunostained with the antibody against olig2 (Supplementary Fig. 13).

Western blot analyses of Sarkosyl-soluble and -insoluble fractions of the frozen normal control and AD brain samples (Brodmann area 6 [BA6], Supplementary Table 3B) revealed that the expression of plexin-B3 was detected only in the Sarkosyl-soluble fractions (Fig. 6m). The specificity of the antibody was further confirmed by western blot (Supplementary Fig. 11B, C). In the Sarkosyl-insoluble fractions of the AD BA6, accumulation of abnormally hyperphosphorylated tau at Ser396 was found to be associated with the Braak stage (Fig. 6m–p). Consistently, total plexin-B3 level was also significantly correlated with the Braak stage (Fig. 6q).

## Discussion

To understand in greater depth the functions of individual cell types in the adult CNS and in disease pathogenesis, homogeneous primary cultures are indispensable. In the present study, we first established a novel reproducible method to purify and culture

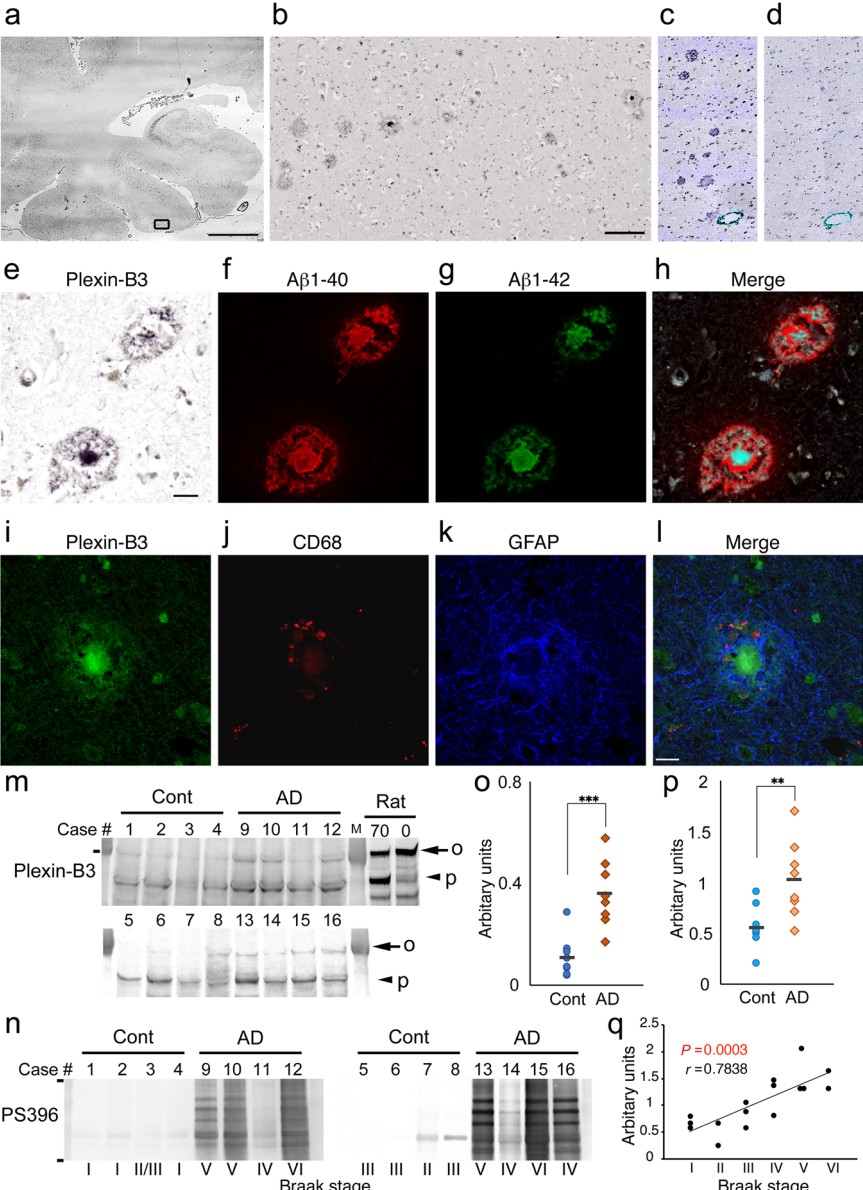

**Fig. 6 Plexin-B3+ senile plaques (SPs) in AD brains. a** Cortical distribution of plexin-B3+ SPs in an AD brain Scale bar: 5 mm. **b** Enlarged image of the rectangular area in **a** (see also Supplementary Fig. 10). Scale bar: 200 μm. **c**, **d** Specificity of the anti-plexin-B3 polyclonal antibody against human plexin-B3 (See also Supplementary Fig. 11). Spatial relationship between plexin-B3+ and Aβ+ areas as analyzed based on the combination of phase contrast (**e**) and confocal images (**f–h**). Scale bar in **d**: 20 μm. See also Supplementary Fig. 12. **i–l** Fluorescent immunostaining of plexin-B3, CD68 (a marker for microglia), and GFAP in an SP. Scale bar: 20 μm. **m** Increased plexin-B3 levels in the Sarkosyl-soluble fractions of AD brains (BA6). Two major bands of plexin-B3 are observed (at 200 kDa, indicated by an arrow and marked "o" and around 130 kDa, indicated by an arrowhead and marked "p"). Note that a postmortem interval increased the intensity of the 130 kDa plexin-B3 band in the rat brains. Rat 0 & Rat 70: adult rat brain homogenate with 0 or 70 h postmortem intervals, respectively. Molecular weight marker: 200 kDa. **n** Accumulation of tau abnormally hyperphosphorylated at Ser396 (PS396) in the Sarkosyl-insoluble fractions of the AD brains. Braak stages (I–VI) are also specified below the blot. Note that abnormal tau accumulation in the AD BA6 is associated with Braak stage. Molecular weight markers: 87 and 35 kDa. **o**, **p** Quantification of the intensity of the plexin-B3 bands in panel **m**. Adjusted arbitrary units were plotted on the Y-axis ($n = 8$/group; **$P < 0.01$, ***$P < 0.001$; unpaired $t$-test). **q** A significant correlation is observed between plexin-B3 levels (arbitrary units; "o" + "p") and Braak stages (Pearson's correlation coefficient: $r = 0.7838$, $P = 0.0003$).

aOPCs from the adult rat brain and discovered an uncharacterized aOPC marker, plexin-B3. We further demonstrated several lines of evidence suggesting that plexin-B3+ aOPCs represent an as-yet-unidentified Aβ-secreting cell type in the adult CNS and most likely in AD. This idea is supported not only by data from the functional analyses of aOPCs in vitro and in vivo, but also by the data from the analyses of human AD brains, especially by the specific plexin-B3 immunoreactivity of SPs, one of the most important neuropathological hallmarks of AD. To the best of our knowledge, the cored SP

pathology has not been observed in any AD animal model, suggesting that simple secretion of Aβs into the extracellular matrix does not sufficiently induce such pathology. Instead, our findings suggest the possibility that Aβs accumulate as cored SPs due to the morphology of the cells secreting them.

Our findings also raised a simple question counter to the conventional amyloid hypothesis, which contends that neurons comprise the dominant and, in fact, sole Aβ-secreting cells in AD. One possible alternative mechanism may involve a type of

demyelination or dysmyelination process in actively myelinating areas of aged gray matter in association with increased Aβ secretion from aOPCs and the formation of cored SPs, thereby promoting axonal dysfunction, neurological disabilities, and most likely formation of neurofibrillary tangles in neurons (Supplementary Fig. 14A, B). This idea is fairly consistent with the neuropathological findings of Braak et al.[12] as well as recent findings from unbiased single-cell transcriptome[13] and spatial transcriptomics[14] analyses of AD brains. If our proposed scenario is true, fine control of cortical (but not white matter) myelination in association with proper APP processing in oligodendrocyte lineage cells in aged brains would be one of the essential requirements of effective AD therapy (Supplementary Fig. 14C); however, the exact roles of APP in aOPCs or Aβs secretion, as well as its processing in oligodendrocyte differentiation remain completely unknown.

We also found considerably unique and as-yet-unidentified APP processing patterns in cultured naïve aOPCs without using genetic engineering techniques such as overexpression (Supplementary Fig. 6C–F). Although the precise mechanisms are still unclear, FGF2 most likely affects both, APP levels and the substrate preference of γ-secretase[47] (Supplementary Fig. 6C–F), as can be partly inferred from the FGF2-dependent band patters of APH-1 and nicastrin (Fig. 3b). Indeed, we found that the secretion of p3-like peptides and intracellular APP-C99 levels increased while APP-C83/APP-C89 levels decreased depending on the concentration of FGF2 (Supplementary Fig. 6C, E). The activities of γ-secretase itself appear to be maintained at all the examined concentrations of FGF2, as cultured aOPCs not only expressed certain levels of γ-secretase components (Fig. 3b, Supplementary Fig. 5E–I) but also secreted NICDs at higher (Fig. 3a) and Aβs at lower concentrations of FGF2 (Fig. 3c, d and Supplementary Fig. 6E). Interestingly, however, Aβx-42/Total Aβ ratios increased ~4-fold at 0 or 1 ng/ml FGF2 (Fig. 3d) when compared to those at higher concentrations of FGF2, suggesting that γ-secretase in aOPCs cultured at 0 or 1 ng/ml FGF2 may be pathogenic by some means. We speculate that similar Aβ secretion from plexin-B3+ aOPCs may also occur in vivo (Fig. 5g, h) in association with the defective oligodendrocyte differentiation, and be involved in the pathogenesis of sporadic AD. Furthermore, specific APP processing, as observed in aOPCs cultured at 0 or 1 ng/ml FGF2 (Fig. 3d and Supplementary Fig. 6F), may be a promising therapeutic target for sporadic AD, although future studies are warranted for validation.

Cortical plexin-B3+ aOPC gliosis induced by CSD (Fig. 5c, d, e) may also represent another promising therapeutic target for sporadic AD if a CSD-type mechanism is found to be involved in the cortical spreading of AD pathology. Very mild CSD stimulation (1 or 2 times during 10 min) (Fig. 5d) was sufficient to induce delayed plexin-B3+ aOPC gliosis in the remote ipsilateral cortex, suggesting that plexin-B3 is one of the most sensitive glial markers for brain injuries. Interestingly, an uncompetitive pan-NMDA-R blocker, memantine, which has been approved for the treatment of dementia, has proven to be partially protective against CSD[48]. Some of its derivatives may also be promising for the amelioration of mild cognitive impairment or early stage AD.

This study also sheds light on several signal transduction pathways, including those involving FGF2 and plexin-B3-semaphorine, as potential therapeutic targets for AD. For example, our in vitro data clearly indicate that the loss of FGF2 signaling is amyloidogenic in aOPCs (Fig. 3). This, however, does not simply imply that FGF2 replenishment confers therapeutic benefits for AD, since it also promotes aOPC proliferation, thereby inhibiting oligodendrocyte differentiation (Fig. 1d, f) and likely disturbing normal myelination. Temporal up- and down-regulation of FGF2 may also be involved in the induction of plexin-B3+ aOPC gliosis

in CSD models[49,50]. Elevated FGF2 levels have also been reported in AD brains[51,52], further highlighting the need for clarifying the exact roles played by FGF2 in the pathogenesis of AD.

In contrast, almost nothing is known about the roles of plexin-B3 in physiological oligodendrocyte differentiation[53] or AD pathogenesis. Expressed mainly in postnatal brains[54], plexin-B3 is a high-affinity receptor specific for semaphorines[27]. Mice lacking plexin-B3 display normal CNS morphology and behavior[54]. Even the fate of plexin-B3+ aOPCs in CSD-induced cortical gliosis remains unclear at present. Since plexin-B3 acts as at least a temporal aOPC marker expressed during the initiation of oligodendrocyte differentiation, plexin-B3+ aOPCs may differentiate into mature oligodendrocytes for the recovery, remodeling, and remyelination of neuronal circuits after cortical injuries. We hypothesize that such mechanisms may be largely disrupted in AD (Supplementary Fig. 14); nevertheless, further studies are clearly needed to understand the fundamental mechanisms of adult cortical myelination, the roles of plexin-B3+ aOPCs in health and disease, the pathophysiology of AD-type de- or dysmyelination, and the possible corresponding therapeutic interventions.

In conclusion, we provide both, a novel culture method for aOPCs and an aOPC marker plexin-B3. We also demonstrate several lines of evidence suggesting that plexin-B3+ aOPCs may be one of the major Aβ-secreting cells in AD. The culture method will be useful for discovering novel functions of aOPCs as well as the regulatory mechanisms underlying adult oligodendrocyte differentiation. Furthermore, these findings may elucidate new AD pharmacotherapies targeting plexin-B3+ aOPC differentiation and cortical (oligodendro)gliosis.

## Methods
All protocols were approved by the Tokyo Metropolitan Institute of Medical Science Animal Care and Use Committee.

**aOPC culture.** Rats (female Sprague-Dawley (SD) rats, >4 months old) were deeply anesthetized with pentobarbital (50 mg/kg). The hippocampus was dissected according to the Paxinos and Watson atlas (http://labs.gaidi.ca/rat-brain-atlas/) (see also Supplementary Table 1[55] for other brain regions), finely minced and digested in papain (26.1 U/ml) (Worthington, Lakewood, NJ) in Hibernate A (BrainBits, Springfield, IL) containing 2% B27 (Thermo Fisher Scientific, Waltham, MA) and 0.5 mM glutamine (Thermo Fisher Scientific) (henceforth referred to as Hibernate A/ B27) for 30 min at 30 °C with shaking at 170 rpm. The tissue was then gently dissociated in 2 ml warm Hibernate A/B27 via brief mechanical trituration; after letting the suspension settle for 2 min, the resulting supernatant (2 ml) was collected. This series of dissociations was repeated thrice; thereafter, the total supernatant (~6 ml) was gently overlaid on a step gradient composed of four parts of 1 ml Hibernate A/ B27 with 16.4% (density 1.041 g/ml), 11.7% (1.029 g/ml), 9.4% (1.023 g/ml), and 7% (1.017 g/ml) Optiprep™ (1.320 g/ml) (Thermo Fisher Scientific)24. After centrifugation at 800 × g for 15 min at room temperature, the top 6 ml of the supernatant was discarded. The remaining ~4 ml consisted mainly of two parts: a large dense band in the middle of the 1.029 g/ml fraction ("sup fraction," or "S" in Figures) and a discrete pellet at the bottom of the tube ("pellet fraction," or "P") (Fig. 1a).

The sup fraction was collected, mixed with 4 ml of medium consisting of Neurobasal A supplemented with 2% B27, 0.5 mM glutamine, 100 IU/ml of penicillin, and 100 μg/ml of streptomycin (Thermo Fisher Scientific) (henceforth referred to as Neurobasal A/B27), and centrifuged at 800 × g for 15 min. The pellet was then resuspended in 10 ml Neurobasal A/B27, and the suspensions were plated onto 60 mm dishes (Corning, Corning, NY) coated with poly-D-lysine (5 mg/50 ml distilled water for 2 h; 135 kDa; Sigma-Aldrich, St. Louis, MO) or onto poly-D-lysine-coated BD BioCoat™ (BD Bioscience, San Jose, CA) chambers. The dishes and chambers were incubated at 37 °C with 5% CO2 for 30–60 min and gently washed to remove unattached cells and debris. Fresh Neurobasal A/B27 containing 20 ng/ml FGF2 (Peprotech, Rocky Hill, NJ) was then added (4 ml to dishes, 400 μl to chambers). Half of the medium was replaced with fresh medium with a double amount of fresh FGF2 added every 2–3 days. The cells were passaged at 70–90% confluence by gentle pipetting. In all the experiments except for that in Fig. 1e (P11), cells passaged up to 7 times were used.

**Immunocytochemistry.** Cells were fixed with 4% paraformaldehyde in PBS for 15 min. After permeabilization with 0.5% Triton X-100 in PBS for 5 min, cells were blocked using 3% bovine serum albumin (BSA) (Sigma-Aldrich) in TBS containing

0.05% Tween 20 (0.05% TBS-T) for 20 min and then incubated with primary antibodies (Supplementary Table 3) overnight at 4 °C. This was followed by incubation with Alexa 488-, Alexa 594-, or Alexa 630-labeled secondary antibodies, TO-PRO-3 (Thermo Fisher Scientific), or Hoechst 33258 in blocking buffer for 60 min.

**Western blot.** For control rat brain samples, female SD rats (>4 month old, unless otherwise indicated) were deeply anesthetized with pentobarbital (50 mg/kg) and placed in a stereotactic frame. For culture samples, aOPCs were washed with cold PBS, collected by scraping, and centrifuged.

The samples (i.e., frozen brain blocks or cell pellets) were lysed in modified-RIPA buffer (50 mM Tris-HCl pH 8.0, 150 mM NaCl, 1% NP-40, 0.25% sodium deoxycholate, 1 mM EGTA pH 7.4) containing a protease inhibitor cocktail (Sigma-Aldrich). After sonication, the lysates were centrifuged at $15,000 \times g$ for 10 min at 4 °C. The resulting supernatants were collected, and protein concentrations were determined using the BCA method (Thermo Fisher Scientific). The samples were then separated using 5–20% gradient SDS-PAGE (Wako, Osaka, Japan) and transferred. The membranes were blocked with 1% BSA (Sigma-Aldrich) in 0.05% TBS-T (blocking buffer) for 1 h, and then incubated overnight with primary antibodies (Supplementary Table 3) in blocking buffer, followed by secondary antibodies conjugated with horseradish peroxidase (HRP) or alkaline phosphatase (Promega, Madison, WI, 1:1,000). Proteins were visualized on a LAS-3000 Mini imager (GE Healthcare, Hino, Japan) using enhanced chemiluminescence (Chemi-Lumi One) (Nakalai, Kyoto, Japan) and a BCIP-NBT Solution Kit (Nakalai).

Quantitative measurements were performed using Multi Gauge V2.3 software (GE Healthcare). In general, measured protein signal was subtracted from the background signal. For the analysis of animal samples, actin or glyceraldehyde-3-phosphate dehydrogenase levels were used for the adjustment of loading bias (Figs. 2d and 3a, b). For the analysis of human samples, the Ponceau S signal was used for the adjustment of loading bias (Fig. 6m–q).

**RNA extraction.** Cultured aOPCs were washed with cold PBS, collected, and frozen at −80 °C until shipping. Total RNA was purified using the RNeasy Mini Kit (QIAGEN, Limburg, Netherlands) in accordance with the manufacturer's instructions. RNA quality was assessed using a Bioanalyzer (Agilent Technologies, Santa Clara, CA).

**Microarray analysis.** Samples were shipped to Agilent Array Services (Hokkaido System Science, Sapporo, Japan). RNA was amplified into cDNA and labeled according to the Agilent One-Color Microarray-Based Gene Expression Analysis Protocol (Agilent Technologies). The samples ($n = 3$) were hybridized to Rat GE 4x44K v3 array slides, and the arrays were then scanned using an Agilent Microarray Scanner (Agilent Technologies). The scanned images were analyzed using the standard procedures described in the Agilent Feature Extraction software 9.5.3.1 (Agilent Technologies).

To compare rat and mouse genes, rat orthologs were checked manually in PubMed (http://www.ncbi.nlm.nih.gov/pubmed) or the Rat Genome Database (http://rgd.mcw.edu/). We excluded a gene from the lists if no rat ortholog was found, or if it was not listed in the Agilent Rat GE 4x44K v3 array. When more than one expression value was obtained for a gene, the largest expression value was used. Complete lists of the cell-type-specific genes are provided in Supplementary Data 1 and 2; complete lists of the top genes are listed in Fig. 1f and Supplementary Fig. 2.

**RNA-seq.** TaKaRa RNA-seq Services (TaKaRa, Kusatsu, Japan) were used for RNA extraction and RNA-seq, including library preparation, fragmentation, and PCR enrichment of target RNA. Samples ($n = 2$) with an RNA integrity number greater than 8 were used for library construction. Sequencing libraries were prepared using a TruSeq RNA Sample Prep Kit (Illumina) according to manufacturer's instructions and then sequenced using the HiSeq 2500 platform (Illumina) to obtain 100 bp paired-end reads.

One hundred bp paired-end reads were aligned to the rat reference genome (University of California, Santa Cruz (UCSC) Genome Browser Jul.2014 (RGSC 6.0/rn6)) using the TopHat tool (version 2.0.14), which incorporates the short-read aligner Bowtie (version 2.2.5). We designated genes with estimated fragments per kilobase of transcript per million mapped reads (FPKM) values had transcripts with an expression of >0 FPKM as "TRUE" and other FPKM values as "FALSE". Complete lists of the cell-type-specific genes are provided in Supplementary Data 3–6, and the top genes are listed in Supplementary Figs. 3 and 4C, E.

**Immunoprecipitation.** The Pierce Classic IP Kit (Thermo Fisher Scientific) was used for Aβ immunoprecipitation. Briefly, conditioned medium (3.6 ml) was adjusted to a 1:1,000 protease inhibitor mixture (Sigma-Aldrich) and then incubated with combinations of antibodies (Aβ1-40, 2 μg or Aβ1-42, 2 μg + APP (22C11, 5 μg) and 20 μl Protein A/G resin overnight at 4 °C.

**ELISA.** A two-site sandwich ELISA[56] was also used for the measurement of Aβ levels. Cultures of embryonic rat hippocampal cells were prepared as previously described[57].

**Image analysis.** Fluorescent images were observed using high-resolution confocal microscopy (Zeiss, Oberkochen, Germany). In some cases, the entire area of the immunostained section was digitalized using a virtual slide system (VS120; Olympus, Tokyo, Japan). Plotting was performed manually on transparent layers overlaid onto the original virtual slide images (×10 objective). For cell counting, defined areas (e.g., Supplementary Fig. 9) were captured using a microscope equipped with AxioCam MRc 5 (×20 objective) (Zeiss).

For quantitative immunofluorescence analysis, cultured cells were imaged and analyzed using the Zen 2011 imaging software (Zeiss). For defining the regions of interest (ROIs), the whole cell territory was first outlined in light blue under the phase contrast image and then lacking areas outlined in yellow were subtracted (an example is shown in Fig. 3f). The total signal intensity of the ROI was measured and adjusted by the area of the ROI (=Mean Intensity).

**Animal studies and brain injuries.** SD rats (Charles River Laboratories, Yoko-hama, Japan) were deeply anesthetized with pentobarbital (50 mg/kg) and placed in a stereotactic frame. For stab wound injuries, 4 male animals (3-month old) were subjected to a stab wound in the right cortex (AP −3.6 mm and ML 2.4 mm from the bregma). Rats were killed at 2 ($n = 3$) and 7 ($n = 3$) days after the injury. For KCl injuries, animals were assigned to the following groups: sham-operated controls ($n = 3$), 2–3 days (2D, $n = 6$), or 6– 7 days (7D, $n = 9$) after 3 M KCl treatment, and 7 days ($n = 3$) after 3 M NaCl treatment. A right parietal trepanation (AP −5.7 mm, L 2.7 mm) was performed and the dura was incised for the topical application of a cotton ball soaked with 3 M KCl or NaCl. After 10 min, the application site was rinsed with saline and the animals were allowed to recover.

To detect proliferating cells in vivo, 10 mg/ml BrdU (50 mg/kg) was administered intraperitoneally to 4 month old female rats twice/day for 5 consecutive days.

For immunohistochemical analysis, the animals were anesthetized and transcardially perfused with PBS followed by 4% paraformaldehyde in PBS. Brains were removed, post-fixed for 1 day at 4 °C, and then cryoprotected overnight in 20% sucrose in PBS. Coronal sections ($n \geq 20$) were prepared using a freezing microtome at 40 μm. Sections ($n \geq 3$) were generally pretreated with DAKO Target Retrieval Solution (pH 9.0) (Agilent Technologies) and then incubated overnight at 4 °C in primary antibodies (Supplementary Table 3) in PBS containing 5% goat serum (or BSA) and 0.4% Triton X-100, and then for 2 h at room temperature in corresponding secondary antibodies with or without TO-PRO-3 or Hoechst 33258. For observation under a light field, the sections were incubated with a streptavidin biotinylated HRP complex (Elite ABC; Vector, Burlingame, CA). Color development was performed using 3,3'-diaminobenzidine (Sigma-Aldrich) in the presence of imidazole and nickel ammonium chloride.

Extracellular direct current field potential recordings were obtained using 0.3-mm-diameter Ag/AgCl microelectrodes. The rinsed recording electrode was positioned on the cortical surface (0.3 mm anterior and 2.7 mm lateral to the bregma) after removal of the dura mater. An Ag/AgCl reference electrode (TN217-002) (Unique Medical, Tokyo, Japan) was placed on the cerebellum surface. Electrodes were connected to a differential headstage, and signals were amplified using a differential extracellular amplifier (ER-1; Cygnus Technology, Delaware, PA; gain of 100). The high cut-off filter was set at 300 Hz. Direct current potentials were digitized at a sampling frequency of 1 kHz with a CED Power1401 data-acquisition system and the Spike2 software (Cambridge Electronic Device, Cambridge, UK).

**Human sample analyses.** This study was approved by the ethics committee of Tokyo Metropolitan Institute of Medical Science. All tissues were collected in accordance with Institutional Review Board-approved guidelines. Permission for autopsy and subsequent use for research was obtained from the next of kin of the patients. Autopsied human brains (Supplementary Table 4A) were fixed in 10% formalin and embedded in paraffin wax; 5–20 μm thick sections were subsequently obtained from the hippocampus. For plexin-B3 immunohistochemistry, serial pretreatment with heat (110 °C in 0.01 M citrate buffer for 10 min) and trypsin (0.05% for 10 min) was performed. Samples were then incubated in primary antibodies diluted in PBS containing 5% BSA and 0.03% Triton X-100 overnight at 4 °C, biotinylated secondary antibodies (1:1000) (Vector) for 2 h. After subsequent incubation with streptavidin biotinylated HRP complex (Vector), color development was performed. For 4G8, Aβ1-42, and plexin-B3 triple immuno(fluoro) labeling, the sections were first immunostained for plexin-B3 as described previously. After the digitalized images were obtained under a light field, the sections were further treated with formic acid (>99% for 1 min) for Aβ double immuno-fluorolabelling (4G8 and Aβ1-42). To investigate the specificity of the sheep polyclonal antibody against plexin-B3 (R&D systems, Minneapolis, MN), the antibody was pretreated overnight with or without 5x recombinant human plexin-B3 (His45-Gln1255 (Gln1156Asp) with a C-terminal 6-His tag) (R&D systems).

For western blot analysis of human brain samples (Supplementary Table 4B), biochemical fractionation of frozen postmortem brain homogenates was performed as previously described[58]. Briefly, unfixed frozen human brain blocks (BA6) were

homogenized in 9 volumes of extraction buffer A68 (10 mM Tris–HCl, pH 7.5, 1 mM EGTA, 10% sucrose, 0.8 M NaCl) containing a protease inhibitor cocktail (Calbiochem, San Diego, CA). After adding 20% (w/v of water) Sarkosyl to concentrations of 2% w/v, the homogenates were left for 30 min at 37 °C, followed by centrifugation for 10 min at 15,000 rpm. The resulting supernatants were further centrifuged at 50,000 rpm for 20 min. The final supernatants were retained as Sarkosyl-soluble fractions, and the pellets were solubilized in PBS as Sarkosyl-insoluble fractions. To examine the effects of postmortem interval on plexin-B3 expression, frozen rat brains dissected the female SD rats (6-month old) at 0 h and 70 h (at 4 °C) postmortem intervals were fractionated and analyzed as described above.

**Statistics and reproducibility**. Statistical analyses were performed using Graph-Pad Prism 8 (La Jolla, CA, USA). For statistical comparisons of two groups, the unpaired t-test was used. For comparisons of more than two groups, analysis of variance (ANOVA) was used followed by post hoc tests. The data are expressed as the mean ± standard deviation unless otherwise indicated.

We characterized the cultured aOPCs using several different methods (i.e., western blot, immunohistochemistry, microarray, RNA-seq). For the investigation of in vitro Aβ secretion, we employed at least two different methods (immunoprecipitation plus western blot [thrice] and ELISA [twice]). In the brain injury study, two different methods (stab wound and KCl injury) were used. In vivo Aβ secretion was found only in the KCl brain injury model at 7D. Anti-plexin-B3 antibody specificity was confirmed using both, western blot and immunohistochemistry. In human brain studies, especially for the biochemical analysis, we employed 8 AD and control brains (BA6) each, and found a correlation between Braak stage and plexin-B3 level in Sarkosyl-soluble fractions

**Reporting summary**. Further information on research design is available in the Nature Research Reporting Summary linked to this article.

## Data availability
The data that support the findings of this study are available from the corresponding author upon reasonable request.

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

## Acknowledgements

We would like to thank T. Tomita for technical advice, and T. Shinozaki, H. Kondo, and A. Nakamura for their technical assistance. This work was supported by JSPS KAKENHI Grant Number 25461794 to N.N.-K. and 15K15438 to Y.T. This research was further supported by AMED under Grant Numbers JP19dm0107105, JP20ek0109392, and JP20ek0109391 to M.Y.

## Author contributions

Y.T. designed the study; N.N.-K., X.-J. Y., T.I., A.T., and Y.T. established the culture model and collected and processed the data; Y.M. and N.O. conducted animal surgeries; N.N.-K. and Y.T. analyzed the animal data; Y.I., A.A., M.Y., S.T., K.H., and T.U. collected human samples; S.T., T.U., N.N.-K., M.H., and Y.T. analyzed the human data; K.S., T.K., and Y.T. analyzed the microarray and RNA-seq data; and Y.T. wrote the paper. All authors discussed the results and commented on the manuscript.

## Competing interests

The authors declare no competing interests.
