## [Peer Review File · Communications Biology]

Reviewers' Comments:

Reviewer #1:

Remarks to the Author:

Neurons and glial cells, such as microglia and astrocytes have been extensively studied for their role in AD pathogenesis. However, the role of oligodendrocyte lineage (OLs) cells in aging and AD remain largely unknown. One of the major obstacles has been the availability of primary cultures from adult rodent brains, especially for OL cells. In the study here, the authors present a novel method to purify adult oligodendrocyte progenitor cells (aOPCs) from rat brains, which can be maintained for several months in defined medium.

Using these cultures, they report the identification of a potential novel marker for aOPCs, namely plexin-B3, which the authors postulate is a delayed marker for cortical gliosis. Moreover, the authors propose a new model, whereby plexin-B3-positive cells secrete Abeta, and suggest that this may be a new mechanism by which Abeta levels increase in AD.

While the findings presented in this study remain tantalizing, there are a number of shortcomings, which significantly decrease enthusiasm for the paper as a whole:

MAJOR ISSUES

1. Study design: There is a significant lack of quantification and statistical analysis in many of the experiments. For example, in figure 2b-c, the authors report that plexin-B3+ cells are "generally NG2-" or "weakly NG2+", but it's not clear at all from the images shown. How is "weakly" defined? Is there an inverse correlation between the levels of NG2 and plexin-B3? The results suggest this may be the case, but without proper quantification, it remains unclear. Other figures either completely lack statistical analysis (e.g. Figure 3d) or both quantification and stats (e.g. Figure 1d, Figure 3b). The number of replicates in most studies are also absent and when mentioned are very low. In the animal studies for example, some groups only include 1 animal (e.g. 3M NaCl 3 and 7 days)! Human studies also include only 4 samples each (Control and AD).

2. Injury model: The relevance of the injury models used here to AD are questionable. While the authors show an increase in the number of plexin-B3 cells, they don't attempt to show any changes in APP processing, similar to what they showed in vitro (i.e. increase APP expression, decrease levels gamma-secretase subunits, increased Abeta secretion, etc), or even changes in tau. As the authors mentioned, brain injuries are a major risk factor for AD, thus it's puzzling why AD-related processes were not analyzed.

3. Immunostaining of human brains: One of the major findings presented here are that plexin-B3+ cells are found in the center of dense-core senile plaques, suggesting that aOPCs are the sources of Abeta. These data are not entirely convincing. Showing for example that plexin-B3+ cells can secrete Abeta in injured brains would add more weight to this hypothesis.

FURTHER QUESTIONS

1. The authors state that these cells can be maintained for months and retain their aOPC properties. It would be good to show microarray or RNAseq data for older cultures.
2. NG2 levels are inversely correlated with plexin-B3. Is depletion of NG2 sufficient to induce differentiation?
3. Notch signaling seems to inhibit OL differentiation. Does gamma-secretase inhibition lead to OL differentiation (e.g. with DAPT)?
4. Levels of gamma-secretase subunits change after differentiation. Does gamma-secretase activity also change? Or target preference (i.e. Notch vs APP? This is an important question that needs to be addressed.
5. Abeta 40 and 42 levels were measured. It would be good to also measure other APP fragments.

MINOR ISSUES

1. Experimental details are missing (e.g. age of rats in rat brain experiments).

2. Other groups have demonstrated roles of other glial cells in AD pathogenesis other than clearance. For example, microglia have been shown to be a major player in synaptic pruning.
3. A few typos are present in the manuscript (e.g. Salkosyl)

Reviewer #2:

Remarks to the Author:

This is an interesting manuscript with novel data; a method for culture of oligodendrocyte progenitors and findings pointing to that these cells can contribute to Abeta production. I think the paper is well written, and presents novel and interesting data that will spark new thinking in the field.

Reviewer #3:

Remarks to the Author:

The paper of Nihonmatsu-Kikuchi describes a novel method to culture adult OL progenitor cells aOPCS. The team made a very detailed characterization of the cultured cells, including IHC, IF, IP, WB, RNA seq, and analyzed published data on transcriptomics to guide their study. By doing this they were able to identify markers of aOPCS and molecular factors that modulate their function, in particular those related to AD amyloidogenesis process. This paper identifies new therapeutic targets and adds significant elements of AD pathophysiology. Evaluating this type of studies in AD mouse models characterizing AD susceptible vs. resistant brain regions would be of even higher relevance.

It is a clear and well design study—I have a general concern, which is the quantification methods—I would like a more detailed explanation, with cut-offs –number of slides, ROIs criteria etc, basically how unbiased was the analysis–i.e. was there a blind evaluation of different fractions preparations?

Specific comments

I would like to have a better understanding of what exactly guided your team to examine the 1029/gr/ml fraction

What does the band pattern change of Aph1 and Nicastrin induced by FGF-2 means? If it is a post-translational modification--do you have a candidate pathway? And is it possible to address it experimentally?

In Fig 3C and methods—what is the rational of using anti-human AB40-42—the lanes seem to have some artifact---please explain

In methods section –preparation of cultures—they did not specify purity of brain region, methods to select it and number and sex of the rats—

Minor comments

In the introduction Line 35-36 is not appropriate because it reads as if they had shown increased secretion of AB42 coming from aOPCS in AD brains—and that is not the case.

Responses to Reviewer 1's comments

1. There is a significant lack of quantification and statistical analysis in many of the experiments.

Thank you for your comment. Accordingly, we added the NG2 and plexin-B3 immunoreactivity quantification data as a new figure (Figure 2c) to better explain the inverse correlation between the levels of NG2 and plexin-B3. We also described the statistical analyses methods used for the data in Figures 1d, 3a, 3b, and 3d. Furthermore, we have increased the number of animal analyses (Figures 5e & f) and included an additional 4 samples each (control and AD); the same changes were found in AD brains (Figures 6m – p). In addition, all data were re-examined when statistical analyses were required (Supplementary Figure 4 & 5). As a result, we believe that we have strengthened our statements.

2. The relevance of the injury models used here to AD are questionable. Show that plexin-B3⁺ cells can secrete A β s in injured brains to add more weight to the hypothesis.

Thank you for your comment. In our extensive analysis of AD-related processes in injured rat brains, we only found that the dot-like A β x-42⁺ structures accumulated extracellularly around some plexin-B3⁺ cells located near the lesions (Figure 5g) at 7 days after the injury (Figure 5h). Considering the specificity of the anti-A β x-42 antibody and the extracellular accumulation of the structures without exception, we believe that this observation is significant. We also believe that the timing and location of the observed changes in the injured rodent brains are important factors, especially as the appearance of plexin-B3⁺ cells itself is not generally pathological, as was demonstrated in the normal adult rat brains.

3. The authors state that these cells can be maintained for months and retain their aOPC properties. It would be good to show microarray or RNAseq data for older cultures.

Thank you for your comment. We confirmed many times (more than 100 times) that these cells could be maintained for months in the defined medium and retained their aOPC properties in terms of olig2 immunocytochemistry. However, with prolonged culture, the expression patterns of some cell-type specific marker proteins, such as NG2, Sox10, and Tuj1, did change (see Figure 1e, P11), mostly due to the increased heterogeneity. We thus stated that, in order to maintain the homogeneity of aOPC culture, we restricted the use of up to 7-passaged cultures. At present, the features of heterogeneity in older cultures are not well characterized. Single-cell RNA-seq, rather than whole culture microarray or RNA-seq, will be most suitable with regard to your comment, but mostly due to the lack of facilities, budget, and man-power, we were unable to actually perform the experiment.

4. NG2 levels are inversely correlated with plexin-B3. Is depletion of NG2 sufficient to induce differentiation?

Thank you for your comment. The role of NG2 in OPC differentiation has been a matter of debate, but remains controversial. This is an important question, and we acknowledge that a thorough investigation is required as an independent study to answer this question. Nevertheless, since we simply used NG2 as a cell-surface marker, answering this question may be beyond the scope and requirements of the present study.

5. Notch signaling seems to inhibit OL differentiation. Does gamma-secretase inhibition lead to OL differentiation (e.g. with DAPT)?

Levels of gamma-secretase subunits change after differentiation. Does gamma-secretase activity also change? Or target preference (i.e. Notch vs APP? This is an important question that needs to be addressed.

Abeta 40 and 42 levels were measured. It would be good to also measure other APP fragments.

Thank you for your comments. Certain levels of gamma-secretase activities seem to be maintained in naïve aOPCs cultured in all FGF2 concentrations examined, since we observed NICD generation at higher FGF2 concentrations (i.e., 20, 40 ng/ml, Figure 3a) and A β secretion at lower (i.e., 0 or 1 ng/ml, Figure 3c & d) concentrations of FGF2. This bidirectional control of processing by gamma-secretase is an unexpected finding, because most cell culture studies regarding Notch and APP processing have shown that when gamma-secretase activities increase, the levels of both Notch and APP processing increase. The main difference between our study and the others is that we used adult CNS-derived naïve cells, while the others mainly used genetically engineered cells, such as those with target protein overexpression. Overexpression is very likely to result in excess amounts of products such as APP-CTFs, which can hamper the interpretation. We even found that APP processing was different between aOPCs and perinatal primary neurons under different concentrations of FGF2 (Supplementary Figure 6C).

There may be two reasons for this bidirectional control: substrate expression and preference. In fact, Notch1 protein levels increased when the FGF2 concentration was high (20, 40 ng/ml), but became negligible when it was low (0, 1 ng/ml). In contrast, APP levels were high when FGF2 concentration was low and vice versa. These changes may result in bidirectional processing. However, we also found some evidence that target (substrate) preference may be involved in FGF2-dependent APP processing. The unique FGF2-dependent APP-CTF pattern (Supplementary Figure 6C & D) as well as inverse appearance of p3-like peptides (Supplementary Figure 6E) both indicate that gamma-secretase prefers APP-C83 as substrates at higher concentrations of FGF2 (i.e., 20, 40 ng/ml), but APP-C99 at lower concentrations of FGF2 (i.e., 0 ng/ml). This unique APP-CTF pattern did not appear in the primary perinatal neuron cultures. Although the precise mechanisms underlying this phenomenon in cultured aOPCs are still unclear, we also found interesting band changes depending on the concentrations of FGF2 for nicastrin and Aph1 (Figure 3b), two components of gamma-secretase likely involved in substrate preference (Yonemura Y. et al. (2016) BBRC 478:1751-1757; Funamoto S. et al. (2013) Nat Com 4:2529).

In this regard, we speculate that pan-gamma-secretase inhibition may not lead to the initiation of OL differentiation. In fact, DAPT treatments induced cell death rather than differentiation at all the FGF2 concentrations mentioned above (data not shown). We have also been trying to use gamma modulators, but the data are not sufficient to draw any conclusions. Therefore, further studies are warranted. We believe that these studies require thorough investigation and should be reported in a separate paper.

6. Experimental details are missing (e.g. age of rats in rat brain experiments).

Thank you for your comment. We carefully re-examined the experimental details and have changed the description accordingly.

7. Other groups have demonstrated roles of other glial cells in AD pathogenesis other than clearance. For example, microglia have been shown to be a major player in synaptic pruning.

Thank you for this comment. The roles of microglia and astrocytes in the pathogenesis of AD remain controversial. In accordance with this comment, we have added additional information in the Introduction.

8. A few typos are present in the manuscript (e.g. Salkosyl).

Thank you for the comment. We have corrected all the typographical errors in the manuscript.

Responses to Reviewer 3's comments

1. Evaluating this type of studies in AD mouse models characterizing AD susceptible vs. resistant brain regions would be of even higher relevance.

Thank you for this comment, which we agree with. However, we think that most AD mouse models, which use neuron-specific promoters such as Thy-1 and CamK-II, may not be useful for

our study, because the genes of interest are expressed only in neurons but not in glial cells in these models. We speculate that, in addition to neurons, aOPCs can also secrete A β s in some pathological conditions in AD. From this point of view, a knock-in mouse model developed by Prof. Saido's Lab might be useful (Saito T et al. (2014) Nat Neurosci 17:661-663). Interestingly, Prof. De Strooper's group has very recently used spatial transcriptomics to investigate the transcriptional changes in tissue domains within a 100 μ m diameter around amyloid plaques in this AD mouse model (Chen WT et al. (2020) Cell 182:976-991); they found early alterations in a gene co-expression network enriched for myelin and oligodendrocyte genes and also confirmed the majority of the observed alterations at the cellular level using in situ sequencing of human brain sections, although the underlying mechanisms are still unclear.

2. I have a general concern, which is the quantification methods; I would like a more detailed explanation, with cut-offs; number of slides, ROIs criteria etc, basically how unbiased was the analysis; i.e. was there a blind evaluation of different fractions preparations?

Thank you for your comment. As suggested, we have added a more detailed explanation of the quantification methods. For example, regarding the ROI criteria in Figure 2c and 3f, we used phase contrast images to determine the boundaries of the cells (an example of ROI definition is shown in Figure 3f). Blind evaluation was performed in counting the cell numbers in Figures 5b, 5e, and 5f.

3. I would like to have a better understanding of what exactly guided your team to examine the 1029/gr/ml fraction.

Thank you for this comment. In the present study, we began our investigation by characterizing the live cells associated with the myelin-debris-enriched fractions since we recognized their presence in our previous studies (Tatebayashi Y et al. (1999) J Neurosci 19:5245-5254). From many published studies on the isolation of microglia and neural precursor cells from adult rat brains and from ours (Tatebayashi Y et al. (1999) J Neurosci 19:5245-5254), it was reasonable for us to anticipate that myelin debris tends to accumulate in the fractions with density gradients spanning 1.029–1.041 (roughly 1.03–1.04), and this was true in this study as well (Figure 1a). As mentioned above, these debris were generally discarded in previous studies.

Accordingly, we have changed the description in the Results section (Section: **Purification and culture of aOPCs from the adult rat brain**).

4. What does the band pattern change of Aph1 and Nicastrin induced by FGF-2 means? If it is a post-translational modification -- do you have a candidate pathway? And is it possible to address it experimentally?

Thank you for this comment. We do not have an exact explanation for the Aph1 and nicastrin band pattern changes induced by FGF-2; however, some studies have suggested that these γ -secretase components may be involved in substrate preference (Yonemura Y. et al. (2016) BBRC 478:1751-1757; Funamoto S. et al. (2013) Nat Com 4:2529), which is why we have mentioned this in the Discussion. We are not aware of any candidate pathways that could have induced such changes; therefore, further studies are warranted.

5. In Fig 3C and methods; what is the rational of using anti-human AB40-42; the lanes seem to have some artifact---please explain

Thank you for this comment. In Figure 3c, we showed human A β 1-40 and A β 1-42 peptides simply as the position markers. Therefore, we cut the lanes in order not to confuse the readers instead added the arrowheads indicating molecular weight, 10 kDa.

6. In methods section; preparation of cultures; they did not specify purity of brain region, methods to select it and number and sex of the rats;

Thank you for this comment. We included the information as suggested, but not in the Methods section, but in the legend of Supplementary Table 1.

7. In the introduction Line 35-36 is not appropriate because it reads as if they had shown increased secretion of A β 42 coming from aOPCs in AD brains; and that is not the case.

Thank you for this comment. We have changed the description accordingly.

(Former: ...Furthermore, we show that plexin-B3⁺ aOPCs probably play roles in the pathogenesis of AD, most likely as natural A β (especially A β 1-42)-secreting cells.)

(New: ...Furthermore, we showed that almost all the SPs in our AD brain samples were immunoreactive for plexin-B3. These data suggest that plexin-B3⁺ aOPCs probably play some role in the pathogenesis of AD, most likely as natural A β -secreting cells.)

Reviewers' Comments:

Reviewer #1:

Remarks to the Author:

The strength of this paper is the development of a potentially useful cell culture model for evaluating the role of adult OL progenitor cells (aOPCs) in Alzheimer's pathogenesis. In its original form, there were several shortcomings which diminished enthusiasm for the paper. The authors have addressed many of these concerns.

1. Lack of quantification and statistical and methodological descriptions: The authors have now addressed the major concerns around their methods and statistics, which strengthen the conclusions.

Remaining issues:

- Description of Fig 2C could be a bit clearer - It's difficult to make out shapes and colors, but that may be due to the low image quality.

- Fig 5f: Iba-1 data is not entirely convincing, as the difference appears to be due to a potential outlier. A proper analysis to rule that out may be warranted. Also, exact p values would be preferable.

- An explanation for how the number of additional animals was determined would also be preferred.

2. Role of gamma-secretase: This remains an important question that the authors don't fully address experimentally. While this may be better left for future studies, a thorough discussion, similar to that provided in the rebuttal may be warranted.

Reviewer #2:

Remarks to the Author:

The authors give extensive and relevant answers to essentially all points raised and provide corresponding data in the revised manuscript.

Reviewer #3:

Remarks to the Author:

All of my comments and suggestions were answered or dealt with appropriately. The manuscript is now clear and is highly informative in the pathogenesis of AD.

Responses to the comments from reviewer 1

- 1. Description of Fig 2C could be a bit clearer – It's difficult to make out shapes and colors, but that may be due to the low image quality.**

We appreciate your observations, and have accordingly checked the resolution of the original figure; we can confirm that it is of high resolution.

- 2. Fig 5f: Iba-1 data is not entirely convincing, as the difference appears to be due to a potential outlier. A proper analysis to rule that out may be warranted. Also, exact p values would be preferable.**

We appreciate your suggestions, and have accordingly rechecked the statistical analysis. We found that even on excluding the highest outlier from the analysis, the numbers of Iba-1⁺ microglia in the 7D CSD group remained significantly higher than those of control group (**P* = 0.024 became **P* = 0.026, unpaired t-test).

- 3. An explanation for how the number of additional animals was determined would also be preferred.**

We appreciate your observations. To keep the number of animals/group at least 3, we performed additional experiments for 7D CSD and 7D NaCl, and added the data as shown in Fig. 5e & f.

- 4. Role of gamma-secretase: This remains an important question that the authors don't fully address experimentally. While this may be better left for future studies, a thorough discussion, similar to that provided in the rebuttal may be warranted.**

We appreciate your observations. The role of gamma-secretase in physiology and disease is a considerably extensive area, and difficult to fully address within the scope of this study. However, we speculate that in the course of oligodendrocyte differentiation, gamma-secretase may play some physiological roles in the quality control of transmembrane proteins. In our aOPC culture system, however, we coincidentally found that defective oligodendrocyte differentiation is associated with certain pathological changes of gamma-secretase and APP processing. We have revised the Discussion to include this information.